# Biomarkers and Neuropsychological Tools in Attention-Deficit/Hyperactivity Disorder: From Subjectivity to Precision Diagnosis

**DOI:** 10.3390/medicina61071211

**Published:** 2025-07-03

**Authors:** Ion Andrei Hurjui, Ruxandra Maria Hurjui, Loredana Liliana Hurjui, Ionela Lacramioara Serban, Irina Dobrin, Mihai Apostu, Romeo Petru Dobrin

**Affiliations:** 1Department of Medicine III, Discipline of Psychiatry, Grigore T. Popa University of Medicine and Pharmacy, 700115 Iasi, Romania; hurjui_ion-andrei@d.umfiasi.ro; 2Department of Morpho-Functional Sciences I, Discipline of Histology, Grigore T. Popa University of Medicine and Pharmacy, 700115 Iasi, Romania; hurjui_ruxandra-maria@d.umfiasi.ro; 3Department of Morpho-Functional Sciences II, Physiology Discipline, Grigore T. Popa University of Medicine and Pharmacy, 700115 Iasi, Romania; ionela.serban@umfiasi.ro; 4Hematology Laboratory, Sf. Spiridon County Clinical Emergency Hospital, 700111 Iasi, Romania; 5Department of Medicine III, Faculty of Medicine, Grigore T. Popa University of Medicine and Pharmacy of Iasi, 16 Universității Street, 700115 Iasi, Romania; irina.dobrin@umfiasi.ro (I.D.); petru.dobrin@umfiasi.ro (R.P.D.); 6Institute of Psychiatry Socola, 36 Bucium Street, 700282 Iasi, Romania; 7Department of Analytical Chemistry, Faculty of Pharmacy, Grigore T. Popa University of Medicine and Pharmacy of Iasi, 16 Universității Street, 700115 Iasi, Romania; mihai.apostu@umfiasi.ro

**Keywords:** attention-deficit hyperactivity disorder, biomarkers, neuroimaging, electroencephalography, cortisol, microbiome, eye-tracking, attention, cognitive dysfunction, quality of life

## Abstract

Attention-deficit/hyperactivity disorder (ADHD) is a prevalent neurodevelopmental disorder with chronic inattention, hyperactivity, and impulsivity and is linked with significant functional impairment. Despite being highly prevalent, diagnosis of ADHD continues to rely on subjective assessment reports of behavior and is often delayed or inaccurate. This review summarizes current advances in biomarkers and neuropsychological tests for the improvement of ADHD diagnosis and treatment. Key biomarkers are neuroimaging methods (e.g., structural and functional MRI), electrophysiological measures (e.g., EEG, ERP), and biochemical measures (e.g., cortisol, vitamin D). Additionally, novel experimental measures, e.g., eye-tracking, pupillometry, and microbiome analysis, hold the promise to be objective and dynamic measures of ADHD symptoms. The review also comments on the impact of the burden of ADHD on quality of life, e.g., emotional well-being, academic achievement, and social functioning. Additionally, differences between individuals, such as age, sex, comorbidities, and the impact of social and family support, are also addressed in relation to ADHD outcomes. In summary, we highlight the potential of these emerging biomarkers and tools to revolutionize ADHD diagnosis and guide personalized treatment strategies. These insights have significant implications for improving patient outcomes.

## 1. Introduction

Attention-deficit/hyperactivity disorder (ADHD) is one of the most common neurodevelopmental disorders, observed in approximately 5–7% of children worldwide and persisting into adulthood in nearly two-thirds of the cases [1,2]. As a highly prevalent disorder with pervasive effects, ADHD presents significant challenges not only to individuals but also to their families, schools, and workplaces. The disorder is characterized by chronic patterns of inattention, hyperactivity, and impulsivity, which usually result in significant cognitive, emotional, academic, occupational, and social impairment [3]. The disorder is also extremely heterogeneous, manifesting differently in different individuals and age groups, which makes the diagnostic process even more challenging [4]. For example, whereas hyperactivity reduces with increasing age, attentional problems tend to persist, making it difficult to diagnose ADHD in adulthood [5].

Despite the growing recognition of ADHD’s impact, a concerning issue remains the diagnosis itself. According to a Brazilian study, among children diagnosed with ADHD, about 35% had little or no impairment in areas of function, calling into question the validity of some of the diagnostic measures and reporting ADHD is frequently overdiagnosed based on symptom checklists alone [6]. This is significant, as overdiagnosis leads to unwarranted intervention and treatment, and underdiagnosis deprives patients of needed support and treatment. The diagnosis of ADHD is largely subjective behavioral assessment, comprising self-report, parent report, and teacher questionnaires, for instance, the Conners’ Rating Scales and the Vanderbilt ADHD Diagnostic Rating Scale [7]. The above, though, has been criticized due to subjectivity and the inconsistency in the reports of the symptoms. Such subjectivity accounts for delays in diagnosis as well as for the failure to discriminate ADHD from other conditions whose symptoms overlap with those of ADHD, such as anxiety, depression, or learning disability [8].

This subjectivity highlights the urgent need for more objective and reliable diagnostic tools. Also, the increasing awareness of the limitations of traditional diagnostic methods has generated significant interest in the development of biomarkers and objective neuropsychological tests that would improve ADHD diagnosis. Recently, a number of studies have investigated neuroimaging, electrophysiological, and biochemical markers as objective measurement techniques for diagnosing ADHD. For example, certain neuroimaging studies documented the existence of atrophic changes within the prefrontal cortex and some of the attention and executive functioning areas, which are usually damaged in children with ADHD [9,10,11]. Research has also reported ADHD patients to have significantly more pronounced structural alteration in the cognitive control network, the dorsolateral prefrontal cortex being a key attentional regulatory area [12].

Furthermore, ADHD can also be diagnosed by using EEG indices such as the theta/beta ratio. These oscillatory processes are ADHD markers, as they reflect attention and cognitive control deficits. For instance, many studies found the ADHD group to have elevated theta/beta ratios, which indicate maintained and concentrated attention. In addition, P300 of event-related potentials is known to be an indicator of cognitive resource allocation/execution speed and attention, and it has been found to be delayed in individuals with ADHD, making it a potential electrophysiological marker of ADHD [13].

In addition to the electrophysiological and neuroimaging markings, biochemical markers have shown importance in identifying ADHD. Certain inflammatory cytokines such as IL-6, IL-10, and TNF-α are suggested to be involved in the pathophysiology of ADHD [14]. A study found that salivary cortisol, a stress biomarker, is proportionately linked to the severity of ADHD symptoms, which suggests a role as a stress biomarker tied to cognitive deficits in ADHD patients [15].

Moreover, new instruments for neuropsychological testing offer an objective assessment of ADHD symptoms. The design of computerized tasks and game-style measures enables the assessment of various cognitive functions, such as working memory, attention, and executive functioning, in real time [16]. This technological advancement allows for more accurate and standardized assessments, helping clinicians to better capture the dynamic nature of ADHD symptoms. Continuous measures of performance through these tasks can enable better standardization and enhanced objective evaluation of ADHD symptoms for clinicians. Continuous real-time monitoring not only provides a longitudinal view instead of a static snapshot but also captures subtle and dynamic dimensions of cognitive activity over time [17,18].

This review intends to offer a general summary concerning the most recent developments in the fields of biomarkers and neuropsychological testing for ADHD diagnosis. In doing so, it will highlight the critical role these innovations play in refining diagnostic processes and guiding treatment. The review will also highlight the role of early diagnosis, the need for improved diagnostic criteria, and how biomarkers and cognitive tests can guide individualized interventions and improved patient outcomes.

This narrative review aims to synthesize existing literature on novel biomarkers and neuropsychological assessment tools used in the diagnosis of Attention-Deficit/Hyperactivity Disorder (ADHD). A comprehensive literature search was performed using two primary electronic databases, PubMed and Google Scholar, covering publications from 2011 to 2025. Search terms included keywords and combinations such as “ADHD diagnosis”, “ADHD biomarkers”, “vitamin D”, “EEG”, “fMRI” and “pupillometry”. All the studies were selected based on their relevance and contribution to the review topic.

## 2. Literature Review

### 2.1. Current Diagnosis of ADHD

The ADHD diagnosis is largely dependent upon behavioral symptoms outlined in the Diagnostic and Statistical Manual of Mental Disorders (DSM-5), which recapitulates symptoms within two categories: inattention and hyperactivity/impulsivity. Though such criteria have been in place for decades, they rely largely on subjective ratings by parents, teachers, and clinicians [19]. Although, for a number of decades, this has been the foundation of ADHD diagnosis, it has some built-in limitations rendering it challenging to measure with validity and reliability.

Parent-teacher rating scales, such as the Conners’ Rating Scales and Vanderbilt ADHD Diagnostic Rating Scales, are typical measures within the assessment process. These tools provide structured formats to measure a child’s behavior in different settings (e.g., home, school). Although very useful for initial screening, they are plagued with high observer bias. The reports are subjective, such that the observation of the behavior is based on experience or expectation and not a reflection of the child’s behavior. This produces false positives (over-diagnosis) and false negatives (under-diagnosis), resulting in delayed treatment or inappropriate intervention. Yet, these scales remain useful triage tools in primary care and other settings where full testing is not always feasible.

#### 2.1.1. Standard Clinical and Behavioral Assessments

The routine diagnostic process typically begins with clinical interviews and behaviour ratings and is supplemented by measures such as the Conners’ Rating Scales and Vanderbilt ADHD Diagnostic Rating Scales. While helpful, they are not free from observer bias, overestimating the occurrence of ADHD. These measures provide a standardized rate of behaviour but remain susceptible to interpretation based on the clinician’s experience. For instance, teacher observation with reference to classroom behaviour is not always an accurate representation of a child’s entire range of symptoms, particularly outside the classroom, for instance, in the home environment or in social environments [20].

The sole major disadvantage of this subjective reliance is that it has the potential to introduce great inconsistency in diagnosis. Studies have proven that even teachers’ and parents’ ratings are not standardized and are dependent on expectations, personal history, and even the environment of observation. This could lead to misdiagnosis and delayed intervention for those children who need it early on [21].

#### 2.1.2. Shortcomings in Existing Methods of Diagnosis of ADHD

While behavioral testing remains the cornerstone of the diagnosis of ADHD, follow-up research has recognized the necessity of objective indicators. Growing awareness of the neurobiological underpinnings of ADHD prompted researchers to seek biomarkers, such as neuroimaging and electrophysiological measurements, to refine diagnostic accuracy and restrict the subjectivity of current measures. Structural and functional MRI studies have revealed patients with ADHD to have atrophic changes in the prefrontal cortex areas, which are areas involved with attention and executive function [9,10,11]. These neurobiological indicators are not yet being employed routinely in routine diagnostic practice, suggesting a widespread deficiency in the field.

Apart from that, the heterogeneity of symptoms of ADHD is also an issue in its diagnosis. The symptoms are highly variable by age, gender, and comorbidities, and hence, it is challenging to utilize a single diagnostic method in every patient. It has been revealed through research that retrospective diagnosis eliminates early symptoms, especially in adolescents and adults, in whom ADHD is overshadowed by other psychiatric disorders such as anxiety and depression [22]. These issues indicate the need for more sensitive diagnostic criteria using both behavioral measures and objective biomarkers.

#### 2.1.3. Importance of Early Screening and Diagnosis

Early detection and treatment are necessary to improve long-term outcomes in children with ADHD. Untreated ADHD is strongly associated with academic underachievement, social impairment, and the development of comorbid conditions such as depression and anxiety [23]. Early interventions, particularly if the symptoms are identified early, can improve academic functioning and social relationships substantially. There is evidence to support the notion that not only do early interventions minimize ADHD symptoms but also lead to more adaptive coping styles, thus improving the quality of life in children with ADHD [24].

Screening initiatives such as the screening program in the UK have decreased waiting times for diagnosis and improved diagnostic efficacy. Waiting times for ADHD diagnosis were reduced from 28 weeks to 12 weeks in a study, which significantly enhanced service provision and patient outcomes [25]. These findings are supportive of adopting effective screening devices and interventions early enough to help avoid long-term social and academic incapacities.

### 2.2. Biomarkers in ADHD Diagnosis

Recent progress in biomarker science has shed light on the neurobiological underpinnings of ADHD and has given rise to the hypothesis that the disorder can be more validly diagnosed by objective biomarkers such as neuroimaging, electrophysiological recordings, and biochemical markers. The subsequent sections describe recent technologies and assess the potential of these technologies to supplement or even supplant traditional subjective assessments, as is illustrated in Figure 1.

#### 2.2.1. Neuroimaging Markers

Neuroimaging techniques, such as MRI, fMRI, and DTI, have provided valuable information regarding structural and functional impairments of ADHD. These techniques have all reported grey matter reduction in the prefrontal cortex and other relevant areas of cognitive control and attention [26,27]. Functional imaging studies have indicated hyperactivity in areas that are critical for attention and response inhibition, especially when cognitive control load is involved in tasks in ADHD patients [28]. These findings attest to the neurobiological basis of ADHD but also suggest the need for further validation before neuroimaging is applied in the clinic.

#### 2.2.2. Electrophysiological Markers

Electrophysiological tests, such as EEG and event-related potentials (ERPs), are another possible domain for the diagnosis of ADHD. EEG studies, particularly those using the theta/beta ratio, have found that ADHD patients exhibit heightened theta activity and reduced beta activity with cognition [29]. The aberrant ratio is thought to reflect the deficit in sustained attention. The P300 ERP component, which reflects the cognitive speed of processing and attention, is also delayed in ADHD patients, further suggesting that electrophysiological tests can be useful diagnostic tools for ADHD [30,31].

#### 2.2.3. Biochemical and Inflammatory Markers

Biochemical markers such as cortisol, vitamin D, and pro-inflammatory cytokines are increasingly emerging as key players in ADHD pathophysiology. Salivary cortisol, as a marker of stress response, has been linked with ADHD symptom severity [6]. Pro-inflammatory markers such as IL-6 and TNF-α are also increased in ADHD patients and may be markers of neuroinflammation [14]. This suggests that biochemical testing can provide additional objective markers to add to current diagnostic protocols.

#### 2.2.4. New Investigational Tools

Along with neuroimaging, other ADHD diagnostic methods such as biochemistry, eye-tracking, pupillometry, and microbiome evaluation are part of emerging diagnostics. These methods open new avenues for diagnosing ADHD-related difficulties, including attention, cognitive control, and physiological responsiveness, which require further assessment and improving clinical approaches to ADHD diagnosis Table 1 summarizes these novel approaches and their diagnostic relevance.

Tracking eye movement is a technology that has been marked as a non-invasive method of assessing interest and cognitive control in ADHD patients. With the help of eye-tracking technology, researchers are able to follow and measure the movements of the eye and the gaze in response to visual stimuli, which indicates the attention span of people with ADHD [32]. Research indicates that ADHD patients experience slower gaze shifts, increased fixation, and frequent lapses in attention relative to control groups. For example, a study found that ADHD patients showed greater-than-expected gaze duration with less-than-expected gaze shift during a visual search task. These results highlight attention deficits that are characteristic of ADHD. Also, eye-tracking has the added advantage of providing objective, real-time, quantifiable information on processes requiring attention, which is very useful in clinical settings where the standard behavioral measures tend to be subjective and dependent on clinician observation [33]. Therefore, tracking eye movement can be useful in the assessment of control over attention in patients with ADHD and reveal the degree as well as the type of attentional deficits.

Pupillometry, or measuring the dilation of pupils in response to cognitive and emotional stimuli, is another laboratory test that is being applied in the ADHD field of research. It was suggested that pupillometry may serve as a measure of cognitive effort or engagement in a task and, therefore, could be an appropriate measure in the investigation of attentional processes in ADHD [34]. Studies have found larger pupil dilations in ADHD patients while performing tasks requiring sustained attention relative to patients without ADHD, suggesting that they are expending more effort to sustain attention. In the study, it was found that children with ADHD had longer pupil dilation for attention-demanding tasks compared to the healthy controls, suggesting children with ADHD exert more cognitive effort relative to the controls. These findings demonstrate the capacity for real-time measurement of attention utilizing pupillometry and the feasibility of using phenomenology as a diagnostic tool for the dysfunctionality of attention in clinical practice [35].

Another very promising area of ADHD research is studying the epidemiology of the microbiome as a possible biomarker. The gut-brain axis has been appreciated more recently for its critical contribution to the development of neurodevelopmental and psychiatric disorders such as ADHD. An imbalance in the gut microbiota accompanying ADHD, known as dysbiosis, is becoming increasingly accepted as a plausible causative factor [36]. An increasing number of studies suggest that children with ADHD are likely to have a different composition of the microbiome than their neurotypical peers. Some species of inflammatory and neurodevelopmental-associated bacteria are known to be overrepresented in ADHD patients [37,38]. For example, Stobernack et al. showed that the gut microbiota of ADHD patients altered after dietary intervention, corresponding to a change in symptom severity. The authors conclude that the ADHD microbiome may respond quickly to some environmental factors and propose that the condition might be influenced by a changeable biological condition rather than a fixed pathology. These results indicate that the microbiome is likely a prospective biomarker of ADHD and that appetite, nutritional interventions, or probiotic treatment may provide strategies for mitigating symptoms [33]. While the relationship between ADHD and the microbiome discussed here suggests a compelling avenue for further study designed to provide clarifying evidence, the possibility of efficacy remains unexplored.

The new biomarkers—eye-tracking, pupillometry, and microbiome analysis—propose to enhance established diagnostic methods and represent a major step towards more objective and individualized ADHD diagnosis. These technologies reveal critical underlying cognitive and physiological ADHD processes and may improve diagnostic precision, particularly in difficult cases where established behavioral assessments are bound to fail. With further research, it is reasonable to assume that ADHD will incorporate these instruments as integral parts of clinical practice, enabling clinicians to deliver more precise, effective, and targeted interventions.

### 2.3. Neuropsychological Assessment and Cognitive Functioning in ADHD

Attention-deficit/hyperactivity disorder (ADHD) is associated with impairments in several cognitive systems, such as attention, executive function, working memory, and response inhibition. These cognitive impairments underpin the clinical picture of ADHD and affect daily functioning in school, social, and work situations. Neuropsychological evaluation is instrumental in diagnosing ADHD and in determining its cognitive substrates. With computerized tasks, gamification, and real-time data tracking becoming increasingly prevalent, tests for ADHD have become increasingly objective, valid, and dynamic.

#### 2.3.1. Primary Cognitive Domains Affected by ADHD

Attention: The hallmark symptom of ADHD is the lack of ability to maintain attention, particularly in tasks that require the ability to maintain attention over time. Research has shown that ADHD patients consistently have deficits in sustained attention and selective attention, leading to compromised performance in tasks requiring the ability to suppress irrelevant stimuli [39]. A meta-analysis showed that children with ADHD made considerably more inattention errors compared to controls in sustained attention tests, including the Continuous Performance Test (CPT) [40].

ADHD Characteristics: Symptoms of attention-deficit hyperactivity disorder (ADHD) include everyday executive functioning skills, which involve goal-setting and action-shifting, planning, organization, and self-regulation (self-control). Among the executive functions overwhelmingly affected by ADHD, inhibition control and working memory seem to be the most relevant ones. Patients with ADHD demonstrate problems with the control of automatic response intervals, as well as the mental process organization [41]. The aforementioned deficits are best demonstrated on the Stroop Test and the Wisconsin Card Sorting Test (WCST), which are two of the numerous tests measuring executive functions [42].

Working Memory: Working memory is another cognitive domain sensitive to tracking ADHD. ADHD individuals usually have deficits in verbal and visual-spatial working memory, which diminishes their ability to perform short information retention and manipulation tasks [43]. Therefore, participants with ADHD were found to perform significantly worse than controls in regard to the Digit Span and Spatial Working Memory tasks [44].

Inhibition: The most profound executive dysfunction related to ADHD is usually regarded as the impairment of response inhibition. This form of impairment is observable in the performance of the Go/No-Go test and Stop-Signal Task (SST), where ADHD patients fail to inhibit prepotent responses [45]. According to Pasini et al., individuals with ADHD, especially those with combined subtypes, have impaired response inhibition that is directly linked to increased impulsivity and hyperactivity [46].

Table 2 shows the primary cognitive domains affected by ADHD, including attention, executive function, working memory and inhibition.

#### 2.3.2. Computer Tasks and Game Elements in Evaluation

The assessment of ADHD has recently been advanced by incorporating computer-based tests and game-like elements to improve the sensitivity of the diagnosis and its ecological validity. Neuropsychological tests that utilize subjective symptoms are far from optimal because, in some cases, they do not assess practical cognitive functioning in real life. However, CANTAB (Cambridge Neuropsychological Test Automated Battery), in its adaptive and dynamic form, allows the measurement of cognitive processes such as working memory, attention, and response inhibition of the ADHD patient, which is often more accurate than manual tests and is often more creative [47].

CANTAB’s components, such as Stockings of Cambridge (SOC), Intra/Extra-Dimensional Shift (IED), and Spatial Working Memory (SWM), reveal differences in the ADHD group compared to controls regarding their cognition [47]. Research has confirmed that ADHD patients performed these tasks significantly worse than controls, supporting the growing evidence of the diagnostic potential of computerized neuropsychological tests for ADHD [48].

In addition to computerized assessments, ADHD symptom measurement through gamification has become increasingly more popular as an interactive strategy, particularly with children. For example, Cogmed Working Memory Training (WMT) is a gamified working memory training task designed to enhance working memory and cognitive control in individuals with ADHD [49]. ADHD children showed improvements in motivation and task performance when their activities were presented in a game format, and not only did they demonstrate better performance on working memory training than in traditional methods, but they also outperformed expectations in many other tasks [50].

#### 2.3.3. Tracking Health Data in Real-Time for ADHD Assessment

ADHD evaluation with real-time data monitoring is indeed a novel, emerging method that tracks cognitive functioning while completing tasks. This method is more precise and accurate for capturing the representation of cognitive functioning and the shifting of attention, working memory, and inhibition on a moment-by-moment basis [51]. Hyperactivity and excess motor activity, which are characteristic symptoms of ADHD, can be measured through movement using a technique known as actigraphy [52]. More recently, researchers have demonstrated that objective measurement of behavior in everyday life through continuous neuropsychological testing can now be supplemented with traditional neuropsychological testing using actigraphy data [40].

Moreover, monitoring cognitive data in real-time can identify the longitudinally emerging cognitive deficits in patients with ADHD; thus, it can improve the discriminative power of diagnosing the condition. Such processes in real-time provide clinicians with better information in regard to the patient’s cognitive profile, enabling the formulation of precise diagnoses and more effective treatment.

## 3. Discussion

ADHD goes beyond simply inattention, hyperactivity, and impulsivity. One of the greatest concerns of ADHD is the diminished quality of life that the affected individual experiences. ADHD impacts many aspects of a person’s life, such as their emotional state, psychosocial milestones, and everyday life activities. With the increasing research focus on ADHD, it has become clearer that assessment of quality of life indicators is critical in evaluating the overall impact of ADHD in patients, more so in children and adolescents.

The emotional well-being of an individual with ADHD is frequently neglected, which weakens self-esteem relative to peers without ADHD. Research chronicled that children with ADHD are at tremendous risk for developing concurrent mood disorders, which disrupt treatment and negatively impact overall prognosis. These emotional concerns are exacerbated by deficits in ADHD-related academic and social functioning, as children with ADHD frequently do not meet hopes academically [1]. As an illustration, Schoeman et al. demonstrate the predominance of bullying and peer rejection among children with ADHD, which embodies a more complicated cycle of impaired emotional control and heightened anxiety [53].

In ADHD, inattention and impulsivity are linked to academic failure. Research shows students with ADHD have difficulties with planning tasks, completing homework, and attending to the class, which manifests as low achievement and school failure [40]. These students may experience worsening life circumstances as a result of the academic struggle, which is most likely a significant predictor of personal satisfaction and social integration.

ADHD has an impact on social life functioning just as it does on academic work. ADHD patients are more likely to be socially impulsive and struggle with self-control in social interactions, resulting in inadequate social skills and peer interaction. Due to their poor social skills, children with ADHD often face peer rejection and struggle in group activities [54]. Fortes et al. highlighted and addressed that children with ADHD reported dissatisfaction in their social relationships, which contributed to heightened anxiety and depression. These concerns could persist into adulthood, in which a person with ADHD would be inefficient in sustaining or nurturing intimate relationships as well as being collaborative and efficient at work [6].

Moreover, the impact of ADHD on daily functioning is profound because it impairs the individual’s ability to perform activities on a daily basis. With ADHD, remembering things such as time, organization, and even managing finances can be overwhelming to people [55]. Individuals with ADHD tend to be more chronically disorganized and exhibit poorer time management skills, resulting in increased, relentless stress and dissatisfaction with daily functioning. Therefore, the effects of ADHD on an individual’s daily functioning are tremendous, transcending the healthcare context and greatly limiting daily activities [23].

To evaluate these different areas of QoL, numerous scales and tools have been designed to assess the multidimensional consequences of ADHD. Some of the most widely adopted measures include WHOQOL-BREF, which evaluates the overall quality of a person’s life in several domains [56]. Another is the ADHD-QOL, which looks into the influence of ADHD in academic, familial, and social interactions [57]. These tools enhance the capacity of clinicians and researchers to make those focused on the individual holistic assessment of ADHD’s impact on their life and subsequently improve treatment plan tailoring and provide more comprehensive individualized care.

### 3.1. Role of Social Support and Individual Differences

Apart from the direct influence of ADHD symptoms on quality of life, individual differences—age, sex, and comorbidities—can influence the effect of the disorder on an individual as well. ADHD is expressed differently across different age groups, with children expressing more overt signs of hyperactivity, whereas adults experience long-term problems with inattention, organization, and impulsivity without the same level of hyperactivity. Research highlighted that adults with ADHD remain undiagnosed since the expression of the disorder decreases with age, resulting in misdiagnosis or delayed diagnosis that makes it difficult to intervene [58].

Neuroimaging biomarkers in children, especially in the context of structural MRI investigations, reveal changes in areas such as the prefrontal cortex, which are related to cognitive control and attention, whereas adults with ADHD reveal less severe structural abnormalities, yet functional connectivity deficits in networks such as the default mode network (DMN) remain. Adult ADHD patients have been shown in functional MRI studies to have ratios of theta/beta for the measurement of attention that are relatively more consistent over time, but the neurological aspects of attention control tend to be less overt in nature than those found in children [13,59].

Another area of the differential biomarkers in children compared to adults involves the regulation of hormones. In children, dopaminergic markers are usually associated with hyperactivity and impulsivity, while adults with ADHD have disrupted neuroendocrine reactions, which have been associated with issues of emotional regulation and insomnia. The age-related differences should be considered when carrying out diagnostic processes to enhance precision among children and adults [60].

Additionally, studies with biochemical markers indicate that adults ADHD indicate a lower correlation between serum biomarkers such as homocysteine and vitamin B12 concentration compared to children. The variations highlight the possibilities of age-related diagnostic biomarkers [61].

Furthermore, gender differences in ADHD symptomatology have come under closer examination over the last several decades. ADHD is widely thought to occur in more males than females; however, research suggests that females may occur in an alternative symptom profile, generally relating to inattention rather than hyperactivity. This results in the underdiagnosis of females because they do not manifest the disruptive behaviors that are the signature of ADHD in their male counterparts [62]. Research states that females with ADHD may use coping strategies that camouflage their symptoms, resulting in difficulty in identification and subsequent diagnosis of the disorder [63,64]. Gender-sensitive diagnostic criteria are thus necessary to ensure that males and females receive appropriate diagnoses and treatment for ADHD.

Comorbidities in ADHD are other factors that are important in understanding the ADHD presentation, which affects the patient’s severity level. The most common comorbidities include anxiety disorders, depression, and learning disabilities that can make the ADHD patient’s academic and social interactions much more difficult [65]. Scientific research discovered that children with ADHD, along with other comorbid anxiety or depression, tend to have greater problems with emotional regulation and a higher risk of underachievement [66]. This means that a thorough, multi-faceted diagnostic process is needed, which includes all the comorbid conditions and detailed treatment plan modifications.

Beyond personal attributes, social support perspectives also help determine the impact ADHD has on the individual’s life in terms of overall well-being. Support from parents, in particular, is one of the strongest protective factors mitigating the symptoms of ADHD. Children with ADHD who receive constant caring support and a well-organized framework from their families academically and socially outperform their peers who receive no such support [67]. The same applies to social contacts. Positive social contacts can mitigate the academic as well as the emotional deficits associated with the disorder; however, negative social contacts in the form of bullying or social rejection exacerbate the emotional vulnerability associated with ADHD [68,69].

### 3.2. Advancements in the ADHD Evaluation and Treatment Processes

ADHD diagnosis and treatment are areas experiencing particularly rapid change, driven by neuroimaging, biomarkers, cognitive profiling, and personalized medicine technologies. The use of objective, non-invasive methods in measuring ADHD, as well as its increasing multidimensionality, offers hope for managing ADHD optimally and improving its lifelong management. While the work is still in progress, the hope is that greater refinement of strategies designed to treat ADHD will result in more individualized and precise approaches, reducing the number of attempts needed to find the optimal treatment.

#### 3.2.1. Implications for Early Intervention and Long-Term Management

Undiagnosed or untreated, ADHD can lead to long-term academic, social, and emotional consequences. However, early intervention can avoid such outcomes by making therapeutic intervention available in a timely manner. The latest research places a strong accent on the role of early intervention programs that focus on behavioral therapy, cognitive skills development, and education, which in turn have been shown to reduce the severity of symptoms as well as improve long-term academic and social functioning [70]. Biomarkers—such as those emerging through neuroimaging and electrophysiology—could significantly raise the precision of early ADHD diagnosis, identifying those at risk before clinically significant symptoms emerge.

A major breakthrough will be personalized therapy for ADHD, apart from a “one-size-fits-all” regimen. As ADHD is a highly heterogeneous disorder, patients will respond differently to various therapeutic interventions based on their individual genetic, neurobiological, and environmental profiles. Greater application of neuroimaging and electrophysiological biomarkers can identify the patients who would most likely benefit from specific types of treatments.

For example, the EEG theta/beta ratio biomarker indicates treatment response to neurofeedback, and other genetic biomarkers predict response to stimulant medications. These treatment strategies might enhance the therapeutic response, reduce adverse effects, and improve overall therapeutic outcomes [13].

#### 3.2.2. Biomarkers and Cognitive Profiling Drive Personalized Interventions

With the progressive conception of ADHD, it is clear that interventions will depend heavily on cognitive profiling as well as biomarker information. Neuroimaging and genetic profiling make it increasingly clear for ADHD to delineate individual differences that manifest within the expression of the condition. For example, patients with ADHD also exhibit attention and executive function problems; some have been shown to possess structural and functional brain lesions such as decreased grey matter within the prefrontal cortex [70]. Understanding the condition from this standpoint enables clinicians to tailor prescriptions along the topic of intervening by directing efforts to very specific neuroanatomical regions or pathways responsible for the symptoms of ADHD the patients present.

Additionally, cognitive profiling—measuring strengths and weaknesses within various domains of an individual’s profile—can inform tailored treatment strategies for the specific needs of each patient. Cognitive assessments such as CANTAB and Cogmed Working Memory Training Diagnostics can pinpoint cognitive impairment, and interventions aimed at constructing executive functions—escalating attention, working memory, and inhibition—can be developed [47,49]. For example, people with severe working memory deficits can be helped by targeted cognitive training, and people with inhibitory control deficits can be helped by neurofeedback-type interventions. The ability to customize treatment based on individual cognitive profiles can enhance the immediate management of symptoms and sustain functional recovery [64].

#### 3.2.3. Preventive Interventions and Early Therapeutic Treatments

In relation to prevention, research is being developed that supports the role of lifestyle changes, including diet, exercise, and sleep, in managing ADHD symptoms. For instance, the DASH diet, along with magnesium and vitamin D, has found some utility in improving ADHD-related behavioral outcomes in children [64]. These nutritional strategies, in combination with behavioral supports, provide an approach for early intervention that eliminates the need for medication, therefore minimizing reliance on medication in cases of mild ADHD.

Furthermore, neurostimulation techniques, such as transcranial magnetic stimulation (TMS) and neurofeedback, are new emerging treatments for ADHD symptoms through modulating specific neural circuits involved in attention and impulse control [71]. These techniques have been shown to potentially induce neural plasticity and improve cognitive functions as a promising adjuvant to drug treatment. However, more research is needed to clarify the long-term effectiveness and most effective application of these treatments, particularly in children [72].

## 4. Recommendations for Future Research

Future research on ADHD must focus on several key areas. First, developmental follow-up studies of the longitudinal course of ADHD symptoms throughout childhood and into adulthood are needed to map out the natural history of the disease and identify its earliest warning signs. These studies could be helpful in identifying the most prognostic long-term outcome biomarkers and the most effective interventions throughout the life course.

Multimodal approaches combining genetic, neuroimaging, and electrophysiological biomarkers with cognitive profiling must be adopted to customize treatment approaches. These approaches will determine the very specific biological and cognitive variables accounting for a particular person’s ADHD in order for more targeted treatments to be formed.

Furthermore, since the science regarding the diagnosis and treatment of ADHD keeps changing, even greater collaboration and teamwork by clinicians, researchers, and policymakers are required to translate new knowledge into clinical practice efficiently. Biomarker research and treatment protocols individually tailored will be of immense value in promoting the quality of treatment of adult ADHD to an even greater extent.

## 5. Conclusions

This review points to the important breakthroughs in comprehending the neurobiological models of ADHD and the encouraging potential of biomarkers and neuropsychological assessment to enhance diagnosis and treatment. While conventional methods of diagnosis are still important, increasing studies of biomarkers herald a transition towards more objective, accurate, and individualized diagnoses. Early intervention with biomarkers and neuropsychological profiling is critical to minimize the long-term effects of ADHD on academic, emotional, and social outcomes.

The incorporation of biomarkers into clinical practice can facilitate personalized treatment, decreasing the trial-and-error process in ADHD care.

Nevertheless, difficulties lie ahead, especially the disparity between research output and its prospective practical implementation. Future research will prioritize longitudinal studies and multimodal integration of biomarkers to better resolve ADHD diagnosis and treatment tactics.

Summing it up, the use of biomarkers and personalized medicine can potentially transform the treatment of ADHD, enhancing the quality of life for patients tremendously by providing more specific and efficient interventions.

## Figures and Tables

**Figure 1 medicina-61-01211-f001:**
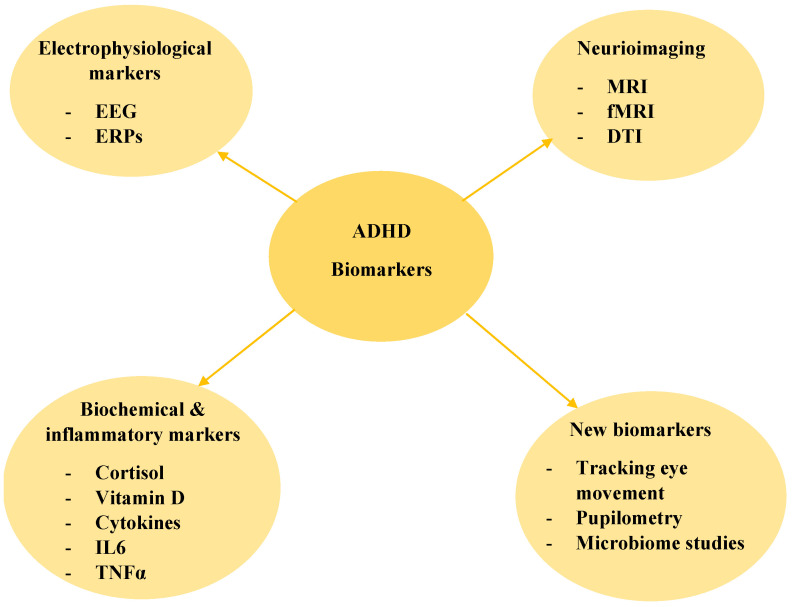
Biomarkers in ADHD Diagnosis: EEG—electroencephalograpy ERPs—event-related potentials, MRI—magnetic resonance imaging, fMRI—functional magnetic resonance imaging, DTI—diffusion tensor imaging.

**Table 1 medicina-61-01211-t001:** Summary of ADHD Biomarkers.

Biomarker Type	Marker	Role in ADHD Diagnosis	Key Findings
Neuroimaging	Structural MRI	Identifies brain structural changes, especially in the prefrontal cortex	Abnormalities in the prefrontal cortex related to cognitive control
	Functional MRI (fMRI)	Measures brain activity during attention and inhibition tasks	Reduced prefrontal cortex activation during attention tasks
	Diffusion Tensor Imaging (DTI)	Evaluate white matter integrity involved in cognitive control	Disruptions in brain tracts associated with attention control
Electrophysiological	EEG (Theta/Beta Ratio)	Assesses brainwave activity related to attention and cognitive deficits	Higher theta/beta ratios in ADHD, indicating attention issues
	Event-Related Potentials (P300)	Measures cognitive processing speed and attention	Delayed P300 latency, suggesting slower cognitive processing
Biochemical	Cortisol	Stress marker linked to ADHD symptoms	Elevated cortisol levels are tied to hyperactivity and impulsivity
	Vitamin D	Associated with brain function and neurotransmitter activity	Low vitamin D levels may affect brain function in ADHD
	Inflammatory Markers (IL-6, TNF-α)	Indicates inflammation, possibly contributing to ADHD	Higher inflammatory markers correlating with symptom severity

**Table 2 medicina-61-01211-t002:** Cognitive Domains Affected by ADHD.

Cognitive Domain	Symptoms/Deficits	Key Tests
Attention	Difficulty maintaining attention, particularly in sustained tasks	Continuous Performance Test (CPT)
Executive Function	Impairments in goal-setting, planning, organization, and self-regulation	Stroop Test, Wisconsin Card Sorting Test (WCST)
Working Memory	Deficits in verbal and visual-spatial memory, affecting retention	Digit Span, Spatial Working Memory tasks
Inhibition	Impairments in response inhibition, leading to impulsivity	Go/No-Go Test, Stop-Signal Task (SST)

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
