# Peer review of "Biomarkers and Neuropsychological Tools in Attention-Deficit/Hyperactivity Disorder: From Subjectivity to Precision Diagnosis"

_medicina, 2025, doi:10.3390/medicina61071211_

Round 1

Reviewer 1 Report

Comments and Suggestions for Authors

4 June 2025

The review on the manuscript, titled ‘Emerging Biomarkers and Neuropsychological Tools in ADHD Diagnosis and Treatment’ by  Hurjui IA, submitted to Medicina

Manuscript ID: medicina-3696675

Dear Authors,

Attention-deficit/hyperactivity disorder (ADHD) is a prevalent neurodevelopmental disorder that manifests through persistent patterns of inattention, hyperactivity, and impulsivity, significantly impairing cognitive, academic, and social functioning across the lifespan. Despite advancements, its diagnosis remains primarily reliant on subjective behavioral assessments, often leading to delayed, inaccurate, or inconsistent identification, highlighting the pressing need for objective biomarkers and neuropsychological tools. In the current manuscript entitled ‘Emerging Biomarkers and Neuropsychological Tools in ADHD Diagnosis and Treatment,’ Hurjui and colleagues investigate emerging biomarkers and neuropsychological tools to enhance the accuracy and objectivity of ADHD diagnosis and treatment.

A key strength of this study lies in its comprehensive synthesis of cutting-edge research on biomarkers and neuropsychological tools for ADHD. It goes beyond traditional behavioral assessments, offering insights into objective, measurable diagnostic methods. By integrating findings from neuroimaging, electrophysiology, and biochemical studies, the authors provide a multidimensional view.

This manuscript presents a timely and relevant investigation that is likely to engage the readership of Medicina. The authors address an important subject; however, the clarity and depth of their arguments would be significantly strengthened by incorporating additional evidence and more comprehensive discussion. I strongly recommend that the authors carefully consider and integrate the provided suggestions to enhance the manuscript’s overall quality.

Comments:

  1. Title: Please provide a concise and informative title that accurately reflects the key message of this study, as this is the most essential aspect of the manuscript. Suggestions: Biomarkers and Neuropsychological Tools in Attention-Deficit/Hyperactivity Disorder: From Subjectivity to Precision Diagnosis; Tracking Minds, Targeting Biomarkers: Advancing Attention-Deficit/Hyperactivity Disorder Diagnosis with Objective Metrics; From Behavior to Biology: Neuroimaging, Biochemical, and Cognitive Markers for Attention-Deficit/Hyperactivity Disorder. Avoid abbreviation in the title.
  2. Abstract:  The abstract is generally clear and coherent; however, a revision following specific guidelines would enhance its structure and impact. It should be limited to 200 words, evenly distributing attention across the background, a concise summary, and a strong conclusion. Begin with one or two sentences introducing the broad research area, followed by two or three sentences that narrow the focus to the specific context and significance. Clearly state the current issue the study seeks to address, and succinctly articulate the research objectives. The problem statement and rationale should be evident and compelling, providing a solid foundation for the study’s importance. Transitioning into the conclusion, begin with a definitive statement summarizing the principal findings—for instance, “Here we highlight”—to underscore the key message. Elaborate on the study’s contributions to the existing body of knowledge, emphasizing its novelty and relevance. Conclude with two or three broader reflections to place the findings within a wider scientific framework, aiming to capture the interest of a diverse academic audience. By refining the abstract in this manner, the authors can create a more balanced, persuasive, and accessible narrative that effectively communicates the significance and broader implications of their research.
  3. Keywords: Please include ten keywords from Medical Subject Headings (MeSH) in the title and the first two sentences of the abstract.
  4. I highly recommend presenting an informative graphical or video abstract.
  5. This section reads well, but a few adjustments could make it even sharper and more engaging. Start broad—paint the big picture to draw readers in—then gradually zoom in on the specific issue your study tackles. Make sure the research gaps are clearly laid out; they should lead naturally to your objectives. Structure the introduction into clear, logical paragraphs, aiming for about 1000 words to allow enough space for developing the main ideas. Keep the language accessible so even readers from different backgrounds can easily grasp the study’s purpose. Guide them carefully from a general overview to a precise statement of what your research sets out to achieve. Throughout, highlight why this work matters by underlining how it fills existing gaps. Above all, maintain a smooth, flowing narrative that connects ideas seamlessly and sets up the rest of the paper in a way that’s both thorough and inviting.
  6. Literature Review: Instead of simply listing studies, pivot to a comparative narrative that pits strengths against weaknesses: show how parent- and teacher-based rating scales inflate prevalence through observer bias yet remain essential triage tools in primary care, and end the passage by flagging exactly where those subjective data still leave diagnostic blind spots
  7. Pull the review into the present by threading in the newest evidence streams—2023-2025 meta-analyses on gut-microbiome shifts, oxidative-stress markers, and real-time eye-tracking—then interrogate how these fresh data complement, contradict, or even upend the better-known fMRI and EEG signatures of ADHD.
  8. Re-engineer the section’s architecture: open with a lean conceptual diagram, march through themed evidence clusters (behavioral, neuroimaging, electrophysiological, biochemical, digital), cap each cluster with a crisp takeaway sentence, and embed a synthesis table or flow-chart for rapid reference; this scaffolding trims repetition, boosts readability, and hands both clinicians and neuroscientists a 1000-word panorama they can actually digest
  9. Discussion: The discussion section is commendably composed; however, to enhance clarity and readability, consider restructuring it into several well-organized paragraphs totaling around 1,500 words without subsections. Begin with an introductory paragraph that sets the stage for the ensuing analysis, and conclude with a summary that encapsulates the key findings from the results. Within the body, develop arguments that underscore the study's primary objectives, the challenges encountered, and the requisite knowledge and technologies to surmount these obstacles. Additionally, provide a general overview of the field, highlighting the significance of your research and its contribution to existing literature. Reflect on the implications of your findings, particularly how they may inform and facilitate future research endeavors. Finally, critically assess the study's strengths and limitations, and discuss potential clinical applications, offering a balanced perspective on the research's impact. Consider incorporating Recommendation for Future Research into this section.
  10. Conclusion: To effectively convey the manuscript's central message, consider incorporating a dedicated paragraph of approximately 150–200 words. This section should highlight the authors' comprehensive and insightful considerations as experts in their respective fields. Emphasizing both theoretical implications and practical applications will underscore the significance of their efforts. Additionally, addressing potential areas for future research and identifying theoretical and methodological aspects requiring further development will provide a more complete understanding of the broader impact and significance of this line of research.

The manuscript contains two figures, no tables, and 82 references. This study offers a substantial contribution to the evolving field of ADHD diagnostics. By critically evaluating emerging biomarkers and neuropsychological tools, it highlights objective methods that surpass traditional subjective assessments. The authors meticulously review neuroimaging, electrophysiological, and biochemical advances, underscoring their clinical relevance. Novel investigational techniques, including eye-tracking and microbiome analysis, are thoughtfully discussed. Moreover, the emphasis on early diagnosis and individualized intervention strategies strengthens its practical implications. Overall, this work provides a rigorous, timely synthesis that propels ADHD research forward. I hope that after careful revision, the manuscript meets the journal’s high standards for publication. In addition, I anticipate the authors preparing “a detailed point-point rebuttal” to my remarks.

Best regards,

Reviewer

Reviewer 2 Report

Comments and Suggestions for Authors

Dear authors,

Thank you very much for the submission.

This review is well structured and emphasizes recent substantial progress in understanding the neurobiological basis of ADHD and points out the promising possibilities of using biomarkers and neuropsychological assessments to improve how ADHD is diagnosed and treated.

1: However, it would be better to have some discussion about the current ADHD biomarker differences between adults and children as well.

2: There is a layout issue on line 559

Round 2

Reviewer 1 Report

Comments and Suggestions for Authors

25 June 2025

The second review on the manuscript, titled ‘Emerging Biomarkers and Neuropsychological Tools in ADHD Diagnosis and Treatment’ by  Hurjui IA, submitted to Medicina

Manuscript ID: medicina-3696675

Dear Authors,

Thank you for your detailed and thoughtful responses to the previous review comments. The revised manuscript has evolved into a well-organized and clearly articulated study that explores promising biomarkers and neuropsychological measures aimed at improving the precision and objectivity of attention-deficit/hyperactivity disorder diagnosis and treatment. I believe the work aligns well with the journal’s standards for publication. I look forward to seeing more contributions from your team in the future.

Thank you.

I have no conflicts of interest to declare regarding this manuscript.

Best regards,

Reviewer

Author Response

Comments from reviewer Thank you for your detailed and thoughtful responses to the previous review comments. The revised manuscript has evolved into a well-organized and clearly articulated study that explores promising biomarkers and neuropsychological measures aimed at improving the precision and objectivity of attention-deficit/hyperactivity disorder diagnosis and treatment. 

Response: We are more than grateful for all the observations and direction gave to us in making final a good manuscript.